# Validation of a Machine Learning Model to Predict Immunotherapy Response in Head and Neck Squamous Cell Carcinoma

**DOI:** 10.3390/cancers16010175

**Published:** 2023-12-29

**Authors:** Andrew Sangho Lee, Cristina Valero, Seong-keun Yoo, Joris L. Vos, Diego Chowell, Luc G. T. Morris

**Affiliations:** 1Head and Neck Service and Immunogenomic Oncology Platform, Department of Surgery, Memorial Sloan Kettering Cancer Center, New York, NY 10065, USA; asl4005@med.cornell.edu (A.S.L.); cvalero@santpau.cat (C.V.); vosj@mskcc.org (J.L.V.); 2Department of Oncological Sciences, Icahn School of Medicine at Mount Sinai, New York, NY 10029, USA; seong-keun.yoo@mssm.edu (S.-k.Y.); diego.chowell@mssm.edu (D.C.)

**Keywords:** machine learning, checkpoint inhibition, head and neck squamous-cell carcinoma, validation, prediction

## Abstract

**Simple Summary:**

Recurrent and/or metastatic head and neck squamous-cell carcinoma (R/M HNSCC) is a clinically challenging disease with a poor prognosis. Despite advances in survival through the use of immune-checkpoint blockade, only a minority of patients experience benefit from such treatments, and it is difficult to identify the patients most likely to benefit. Machine learning approaches integrating clinical and genomic data can predict response to immune-checkpoint blockade across all cancers; however, the performance of this model in HNSCC has not been examined. Here, we validate this previously described immune-checkpoint blockade response prediction model in R/M HNSCC patients. This model was able to predict response as well as overall survival following immune-checkpoint blockade in patients with R/M HNSCC. Further investigation will be needed to further delineate the importance of HNSCC-specific features.

**Abstract:**

Head and neck squamous-cell carcinoma (HNSCC) is a disease with a generally poor prognosis; half of treated patients eventually develop recurrent and/or metastatic (R/M) disease. Patients with R/M HNSCC generally have incurable disease with a median survival of 10 to 15 months. Although immune-checkpoint blockade (ICB) has improved outcomes in patients with R/M HNSCC, identifying patients who are likely to benefit from ICB remains a challenge. Biomarkers in current clinical use include tumor mutational burden and immunohistochemistry for programmed death-ligand 1, both of which have only modest predictive power. Machine learning (ML) has the potential to aid in clinical decision-making as an approach to estimate a tumor’s likelihood of response or a patient’s likelihood of experiencing clinical benefit from therapies such as ICB. Previously, we described a random forest ML model that had value in predicting ICB response using 11 or 16 clinical, laboratory, and genomic features in a pan-cancer development cohort. However, its applicability to certain cancer types, such as HNSCC, has been unknown, due to a lack of cancer-type-specific validation. Here, we present the first validation of a random forest ML tool to predict the likelihood of ICB response in patients with R/M HNSCC. The tool had adequate predictive power for tumor response (area under the receiver operating characteristic curve = 0.65) and was able to stratify patients by overall (HR = 0.53 [95% CI 0.29–0.99], *p* = 0.045) and progression-free (HR = 0.49 [95% CI 0.27–0.87], *p* = 0.016) survival. The overall accuracy was 0.72. Our study validates an ML predictor in HNSCC, demonstrating promising performance in a novel cohort of patients. Further studies are needed to validate the generalizability of this algorithm in larger patient samples from additional multi-institutional contexts.

## 1. Introduction

Head and neck squamous-cell carcinoma (HNSCC) is a group of cancers originating from epithelial cells lining the mucosa of the upper aerodigestive tract [1]. Most patients present at an advanced stage at first diagnosis. The survival rate of HNSCC patients has only seen modest improvements for decades [2]. Accordingly, HNSCC represents the seventh most common cause of cancer death in the world, causing around 450,000 deaths annually [1,2]. Additionally, about half of treated patients experience recurrent and/or metastatic (R/M) disease. R/M HNSCC is generally incurable, with a poor prognosis and median overall survival ranging from 10 to 15 months [3]. Treatment of R/M HNSCC presents a difficult ongoing clinical challenge.

Checkpoint inhibitor immunotherapy is emerging as an important modality for the treatment of R/M HNSCC. Immune-checkpoint inhibitors, or immune-checkpoint blockade (ICB) drugs, target signaling molecules on the surface of immune and tumor cells to overcome escape and enhance immune killing [4]. Indeed, ICB drugs have improved survival for patients with HNSCC, compared to the previous standard of care. However, most patients treated with ICB do not experience clinical benefit and are subjected to treatment-related adverse effects and possibly financial toxicity [5,6]. There is an unmet clinical need for greater precision in the use of ICB drugs through the identification of patients who are most likely to benefit, as well as the identification of patients who are unlikely to experience benefit, for whom other therapies (such as cytotoxic chemotherapies) may offer greater clinical benefit. Although predictive biomarkers have been proposed, such as tumor mutational burden (TMB) and immunohistochemistry for programmed death-ligand 1 (PD-L1), these biomarkers have shown poor-to-modest predictive value, and no single biomarker has been able to predict response with high accuracy: TMB and PDL1 individually achieved area under the receiver operating curve (AUROC) values of 0.61 and 0.62, respectively in R/M HNSCC patients [7,8,9].

With the goal of developing more precise predictive tools, machine learning algorithms have been employed, making use of nonlinear relationships between multiple variables to achieve greater predictive ability than one biomarker alone. Previous machine learning approaches have been used to make predictions from radiomic features [10,11], tumor microenvironment gene signatures [12,13], and hematoxylin and eosin images [14]. Machine learning has also been used to weakly predict immune-related adverse effects from immunotherapy, which currently lacks early-phase biomarkers [15,16]. However, previous studies have been limited by small sample size and have not been validated, limiting their clinical utility [17]. Chowell and colleagues successfully developed a pan-cancer prediction model for ICB response using an array of clinical, genomic, and laboratory features [18]. They used a random forest (RF) machine learning classifier trained and tested on a pan-cancer cohort, which was divided into three groups due to patient data availability: melanoma, non-small-cell lung cancer, and other cancer types. The “others” group included 14 different cancer types, including HNSCC, which accounted for 55 patients out of 1184 patients in the training set. Although the group as a whole performed well, the performance of the entire group may not be indicative of performance in each cancer type; each cancer type in the “others” group represents a small fraction of the entire training cohort. Moreover, although some of the information between input features is relevant across cancers, this may not outweigh type-specific differences in every cancer type. Therefore, performance on a multi-cancer cohort is not enough to guarantee efficacy in HNSCC patients, and validation on a separate patient cohort is necessary.

Here, we examine the performance of the previously described pan-cancer RF models in an expanded HNSCC-only validation cohort. We report the predictive performance of the RF classifiers by measuring the area under the receiver operating characteristic curve and the precision–recall curve, assess the calibration of the predictions, and assess the ability of the model to stratify survival outcomes in the HNSCC validation cohort.

## 2. Materials and Methods

### 2.1. Patient Cohort

Patients with recurrent/metastatic HNSCC who had received at least 1 dose of anti-PD-1, anti-PD-L1, or anti-CTLA-4 ICB at Memorial Sloan Kettering Cancer Center (MSKCC) from 2015 to 2019 were retrospectively identified. All patients had tumor DNA profiling as part of their routine clinical care using MSK-IMPACT next-generation sequencing [19]. Patients who received neoadjuvant ICB (*n* = 9), patients with cutaneous squamous-cell carcinoma (SCC) (*n* = 2) or histology other than SCC (*n* = 17), patients with synchronous primary tumors (*n* = 3), and patients with incomplete data (*n* = 13) were excluded. Of these patients, data from 55 patients had been previously used in the development of the model by Chowell et al., and these patients were therefore excluded from the validation cohort. In total, 96 patients with complete clinical and tumor sequencing data were identified. Patient data use was approved by the MSKCC Institutional Review Board (IRB), and all patients provided informed consent. This cohort of HNSCC patients was distinct from the cohort used for the training and initial testing of the model in the original study by Chowell et al. [18].

### 2.2. Clinical Outcomes

The study outcomes were response to ICB, overall survival (OS), and progression-free survival (PFS). Response was categorized based on the Response Evaluation Criteria in Solid Tumors (RECIST) v1.1 [20]. For patients without formal RECIST reads (*n* = 44), physician notes were reviewed manually and tumor response was categorized using the same criteria. Patients who exhibited complete response (CR) and partial response (PR) were classified as responders (R); patients who experienced stable disease (SD) or progressive disease (PD) were classified as non-responders (NR). Clinical benefit was defined as an objective response or stable disease lasting at least 6 months. PFS was defined as the time from the first ICB infusion to disease progression or death from any cause; patients without progression were censored at their time of last contact. OS was defined as the time from the first ICB infusion to death from any cause; patients who were alive at the time of review were censored at their time of last contact.

### 2.3. Genomic, Demographic, Molecular, and Clinical Data

For each patient, the data collected included age, sex, body mass index (BMI), tumor stage at time of first ICB infusion, immunotherapy drug agent, and whether the patient received chemotherapy before immunotherapy. BMI was calculated by dividing the patients’ body weight (kg) by the square of their height (m^2^) before ICB treatment. Tumor stage was determined according to the American Joint Committee on Cancer, 8th edition [21].

Peripheral blood values were obtained from the closest blood test before the first ICB infusion collected for routine laboratory testing, all within 1 month before the start of ICB. Laboratory data included blood albumin in g dL^−1^, platelets per nanoliter, hemoglobin (HGB) in g dL^−1^, and neutrophil-to-lymphocyte ratio (NLR), defined as the absolute count of neutrophils (per nanoliter) divided by the absolute count of lymphocytes (per nanoliter).

Genomic data included tumor mutational burden (TMB), fraction of genome with copy number alteration (FCNA), HLA-I evolutionary divergence (HED), loss of heterozygosity (LOH) status at the HLA-I locus, and microsatellite instability (MSI) status. TMB was defined as the total number of somatic tumor non-synonymous mutations normalized to the exonic coverage in megabases. FCNA and HLA-I LOH allelic status were determined using allele-specific copy number analysis [22]. MSI status was calculated using MSIsensor [23] with the following criteria: stable (0 ≤ MSI score < 3), indeterminate (3 ≤ MSI score < 10), and unstable (10 ≤ MSI score). We dichotomized groups by MSI status: MSI unstable versus MSI stable/indeterminate. HED was calculated as previously described by Chowell et al. [18] by obtaining the protein sequence of each allele of each patient’s HLA-I genotype and determining the mean of divergences calculated by the Grantham distance metric at HLA-A, HLA-B, and HLA-C.

Features were encoded as described previously [18]. Because the goal of this analysis was to validate the previously described model, the features from the original model were not modified. Continuous features included age, BMI, albumin, platelets, HGB, NLR, TMB, FCNA, and HED. Categorical features, which were encoded as binary values, included sex (male or female), tumor stage (stage IV or I–III), chemotherapy prior to ICB (yes or no), drug class (combo or monotherapy), HLA-I LOH (yes or no), and MSI (unstable or stable/indeterminate). Cancer type was encoded as “Others”.

### 2.4. Application of the Random Forest Classifier

We tested the 2 RF models previously described by Chowell et al.: the RF11 and RF16 models. The RF11 model uses 11 input features associated with ICB efficacy to provide a score predicting ICB response. The features included in RF11 are age, sex, BMI, tumor stage, drug class, NLR, TMB, FCNA, HED, HLA-I LOH, and MSI. The RF16 model includes all of the features in RF11, in addition to albumin, chemotherapy prior to ICB, platelets, HGB, and cancer type. A higher RF prediction score indicates a higher chance of response to ICB. Using the published RF11 and RF16 models, with identical hyperparameters, we obtained the RF prediction scores for our cohort.

### 2.5. Performance Metrics and Statistical Analysis

Patient characteristics and outcome data in the validation and development cohorts were compared using chi-squared test for proportions and the Wilcoxon signed-rank test for continuous variables.

The primary outcome for the study was binary discrimination, or the model’s ability to predict responders and non-responders as measured by the area under the receiver operating characteristic curve as well as the area under the precision–recall curve (AUPRC). An optimal cutoff point was determined in the development cohort by maximizing Youden’s index and then used to make discrete risk-group classifications in the validation cohort, from which we report sensitivity, specificity, positive predictive value (PPV), and negative predictive value (NPV) using the pROC package [24]. Metrics and 95% confidence intervals were computed with 2000 stratified bootstrap replicates.

Calibration of the model reflects the difference between the predicted and observed response probabilities [25]. First, to assess the mean calibration, we compared the average predicted scores across all patients with the overall response rate. To examine the quality of the model probabilities, we used a calibration plot by dividing the predictions into equally sized bins and constructing the plot from the average predicted probability of each bin and the observed proportion of responders. Additionally, the intercept and slope were calculated, representing calibration-in-the-large and the calibration slope [26]. Perfectly calibrated plots should be on the ideal line, with an intercept of 0 and a slope of 1. Calibration plots and metrics were obtained using the CalibrationCurves package [27].

We tested the model’s ability to predict OS and PFS in patients treated with ICB by calculating Cox proportional hazard ratios and evaluating Kaplan–Meier survival plots. Log-rank *p*-values were calculated using the survminer package. We followed the 2015 guidelines for Transparent Reporting of a Multivariable Prediction Model for Individual Prognosis Or Diagnosis (TRIPOD): Prediction Model Validation Checklist [28] (Appendix A).

## 3. Results

### 3.1. Population Characteristics

The clinical characteristics of 96 patients in the validation cohort consisting of R/M HNSCC patients and the pan-cancer development cohort are shown in Table 1. The proportion of male patients was higher in the validation cohort compared to the complete development cohort (79% vs. 55%). A greater proportion were also categorized as stage IV at the time of ICB in the validation cohort compared to the development cohort (100% vs. 93%) and had been treated with chemotherapy (96% vs. 69%). Additionally, a greater proportion of patients in the validation cohort were treated with PD-1/PD-L1 inhibitors compared to the development cohort (93% vs. 83%). Importantly, the ICB response rates were not significantly different between the cohorts (25% vs. 28%). The characteristics of the HNSCC patients who were only in the training set of the development cohort are also shown—these patients had no significant differences from the validation cohort.

### 3.2. Validation

The RF11 and RF16 classifiers were used to generate ICB response probabilities for 96 R/M HNSCC patients using their clinical, laboratory, and tumor genomic features (Figure 1A). For each model, the importance of each feature was calculated by determining the change in the model when the feature was reduced to random noise (Appendix A). RF11 and RF16 achieved moderate AUROC (RF11 AUROC = 0.65 [95% CI 0.53–0.77], RF16 AUROC = 0.60 [95% CI 0.46–0.74], TMB AUROC = 0.57 [95% CI 0.44–0.71]) (Figure 2A). Additionally, both RF11 and RF16 showed higher AUPRC compared to TMB (RF11 AUPRC = 0.38 [95% CI 0.26–0.55], RF16 AUPRC = 0.39 [95% CI 0.25–0.55], TMB AUPRC = 0.28 [95% CI 0.21–0.39]); the baseline AUPRC was 0.26 (Figure 2B).

Using the patient data of HNSCC patients (*n* = 55) in the training set of the original development cohort, we found the optimal cutoff point of 0.26 for the RF11 score, which maximized Youden’s index (Figure 1B). Similarly, the optimal cutoff point for the RF16 score was 0.24.

We stratified the cohort into predicted responder and non-responder groups; patients with response probabilities above or below the cutoff point were classified into the “predicted responder” and “predicted non-responder” groups, respectively. We report sensitivity, specificity, PPV, NPV, accuracy, and F_1_ score (Table 2). The RF11 model had overall accuracy of 0.72, while the RF16 model had overall accuracy of 0.69. The RF11 model had an F_1_ score of 0.34, and the RF16 model had an F_1_ score of 0.31.

Mean and weak calibration was assessed. Overall, the mean RF11 was 0.22 and the mean RF16 was 0.19, compared to the actual response rate of 0.25. The calibration plots demonstrate that the models’ predictions show adequate weak calibration, as assessed by the calibration slope and intercept. For RF11, the calibration slope was 1.98 (95% CI 0.28–3.68) and the calibration intercept was 0.17 (95% CI −0.29–0.64) (Figure 3A). Likewise, for RF16, the calibration slope was 1.10 (95% CI −0.06–2.27) and the calibration intercept was 0.33 (95% CI −0.14–0.80) (Figure 3B).

The univariate Cox model using the response probability classifications found that the patients classified as predicted responders by RF11 had significantly improved OS and PFS. The hazard ratio for overall survival for the “predicted responder” group was 0.53 (95% CI 0.29–0.99, *p* = 0.045); for PFS, the hazard ratio was 0.49 (95% CI 0.27–0.87, *p* = 0.016) (Figure 4A). However, patients classified as predicted responders by RF16 did not have differences in OS and PFS (Figure 4B). The hazard ratio for RF16-classified predicted responders for OS was 0.70 (95% CI 0.40–1.21, *p* = 0.2) (Figure 4C); likewise, the hazard ratio for PFS was 0.83 (95% CI 0.50–1.40, *p* = 0.48) (Figure 4D).

## 4. Discussion

In this study, two previously developed prediction models for estimating the probability of ICB response in all cancer types were validated in a cohort consisting exclusively of head and neck cancer patients. In the HNSCC cohort, both the RF16 and RF11 models had moderate discriminative performance in identifying possible responders and non-responders, along with reliable risk percentages.

The development cohort’s performance was measured with a test set consisting of 295 patients [18]. This set was grouped according to cancer type—non-small-cell lung cancer, melanoma, and others—and the performance was assessed by group. The “Others” group contained 150 patients, 14 of whom were head and neck cancer patients. The RF16 achieved an AUROC of 0.79 in the entire test set and the “Others” group in the development study [18]; however, isolating only the HNSCC patients in the test set, the RF16 achieved an AUROC of 0.52 compared to the training set’s AUROC of 0.97, suggesting limited or unknown generalizability of the model for HNSCC patients. When tested on the validation cohort, RF11 achieved an AUROC of 0.65 and RF16 achieved an AUROC of 0.60, both lower than in the pan-cancer test set, but greater when assessing HNSCC patients exclusively in the test set.

Interestingly, RF11, which excludes five input features from RF16, had improved performance, with an AUROC of 0.65 and superior ability to predict survival outcomes. One or more of the excluded features may have been informative in a pan-cancer model but not informative in the context of head and neck cancer, resulting in better performance upon their removal. Indeed, when features’ importance was computed for RF16, the removal of platelets as a feature improved the performance of the model. However, the fact that this feature is unimportant in this model does not preclude its importance in other models; a future model with high predictive performance may be able to utilize this feature.

The model’s response prediction probabilities adequately estimated the observed response rates, with a trend to underestimate risk, as indicated by the mean calibration, calibration plot, slope, and intercept. However, the small sample size of this validation cohort should be taken into consideration in the interpretation of the calibration. Specifically, in the higher risk percentages, the small population of each bin limits the assessment of calibration at those ranges. Therefore, only mean and weak calibration were able to be assessed; future studies may provide additional data to evaluate moderate and strong calibration using a flexible calibration curve [29]. Taken together, our results demonstrate that the pan-cancer prediction model had decreased performance in a specific cancer type that was underrepresented during its development. Information from features and their relationships may not have universally consistent meaning; therefore, future predictors could make use of more data to develop and validate in each cancer type. Our results have several limitations. First, the patient cohort was from the same institution and time period as the development cohort; thus, this study does not assess the predictor’s geographic or temporal validity. Our patient cohort may not be representative of the global HNSCC patient population, and the feature data, especially genomic data, may be obtained and calculated differently in other institutions. Therefore, future application of the predictor in different settings will be required to assess reproducibility. Additionally, the sample size of the validation cohort was restricted by the number of patients that fit the inclusion criteria in a single institution, limiting the precision of the performance estimates. Future studies could integrate data from multiple institutions for an improved validation of the model or develop a new prediction model altogether, including more features that are known to impact prognosis in HNSCC, such as smoking or HPV status [30].

## 5. Conclusions

This study validates a previously developed ICB response prediction model in a cohort of immunotherapy-treated HNSCC patients. We demonstrate adequate discriminatory performance and well-calibrated response probabilities. This model uses clinical, genomic, and laboratory data that are routinely collected at our institution and has the potential to aid in clinical decision-making and patient risk stratification. As the availability of data improves and new markers of tumor immunology are discovered, similar prediction models are expected to improve upon this work to assist in providing accurate, personalized therapy.

## Figures and Tables

**Figure 1 cancers-16-00175-f001:**
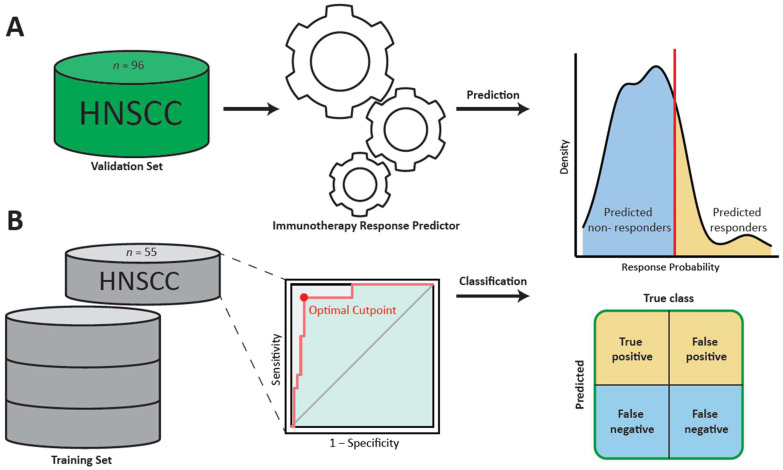
Schematic representation of the validation of the prediction model: (**A**) The random forest predictors were used to obtain response probabilities for each patient in the validation set. (**B**) The data of HNSCC patients in the training set of the development cohort used by Chowell et al. were used to determine a cutoff point for response probability. Finally, this cutoff point would be used in the validation to make discrete classifications of predicted response (HNSCC: head and neck squamous-cell carcinoma).

**Figure 2 cancers-16-00175-f002:**
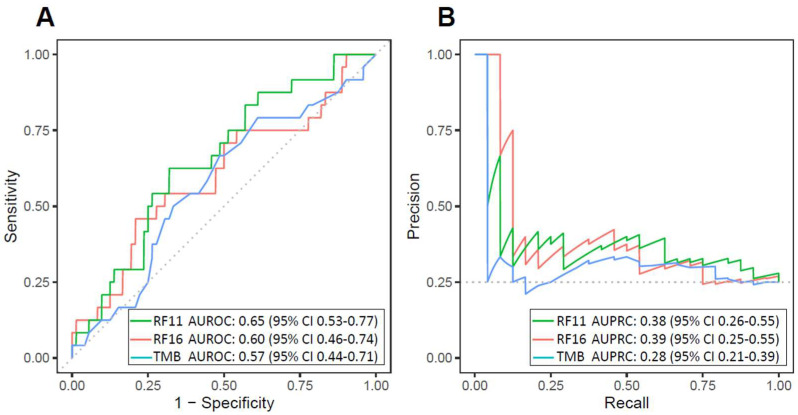
Discrimination of head and neck squamous-cell carcinoma immune-checkpoint blockade response by the 11-feature random forest model (RF11) and the 16-feature random forest model (RF16) in the validation cohort: (**A**) Receiver operating characteristic curves and (**B**) precision–recall curves and corresponding areas under the curves for RF11, RF16, and TMB. The baseline AUPRC is 0.26 (AUROC: area under the receiver operating characteristic curve; AUPRC: area under the precision–recall curve).

**Figure 3 cancers-16-00175-f003:**
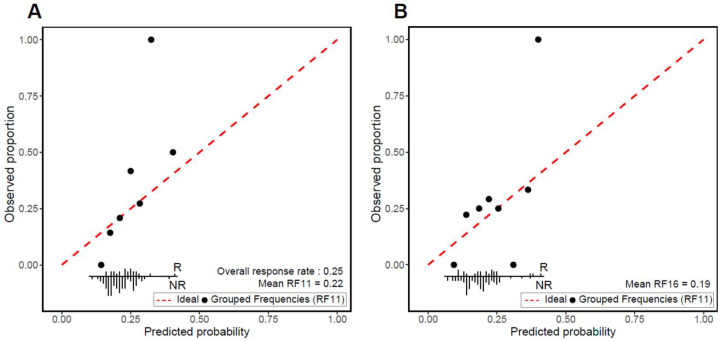
Calibration plots for (**A**) RF11 and (**B**) RF16 immune-checkpoint blockade (ICB) classifiers. Observed proportions of ICB responders and mean predicted probabilities were compared across deciles of predicted probabilities. The red dotted line denotes perfect agreement between the predicted and observed response. Frequency distribution according to outcome is shown at bottom of each plot (R: responder; NR: non-responder).

**Figure 4 cancers-16-00175-f004:**
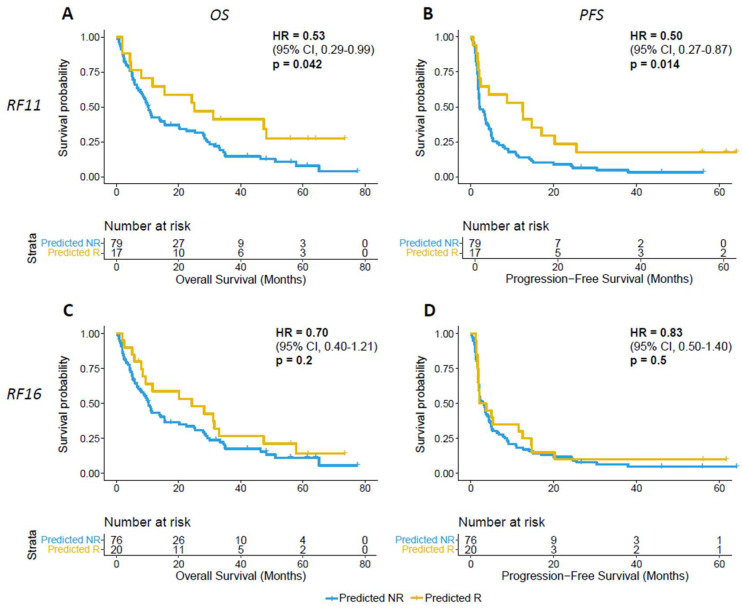
(**A**,**B**) RF11 and (**C**,**D**) RF16 prediction of ICB association with overall survival (OS) and progression-free survival (PFS) (R: predicted responder; NR: predicted non-responder. Two-sided *p*-values were computed using the log-rank test).

**Table 1 cancers-16-00175-t001:** Baseline patient characteristics and outcomes between the development cohort used by Chowell et al. [18] and the validation cohort.

Characteristics and Outcomes	Validation Cohort (*n* = 96)	HNSCC Patients in Development Cohort, Training Set (*n* = 55)	Development Cohort (*n* = 1479)
Age, median, years (IQR)	62 (54–69)	62 (52–67)	64 (55–71)
Sex, n (%)			
Female	20 (21)	14 (26)	668 (45.2)
Male	76 (79)	41 (74)	811 (54.8)
Cancer type, n (%)			
Head and neck	96 (100)	55 (100)	69 (4.67)
Stage, n (%)			
I–III	0 (0)	1 (2)	97 (6.6)
IV	96 (100)	54 (98)	1382 (93.4)
Chemotherapy prior to ICB, n (%)			
Yes	92 (96)	51 (93)	1016 (68.7)
No	4 (4)	4 (7)	463 (31.3)
Drug class, n (%)			
PD-1/PD-L1	67 (93)	53 (96)	1221 (82.6)
CTLA-4	0 (0)	1 (2)	5 (0.3)
Combo	7 (7)	1 (2)	253 (17.1)
ICB response, n (%)			
Yes	24 (25)	11 (20)	409 (27.6)
No	72 (75)	44 (80)	1070 (72.4)

IQR: interquartile range, ICB: immune-checkpoint blockade, Combo: PD-1/PD-L1 and CTLA-4.

**Table 2 cancers-16-00175-t002:** Classification metrics for HNSCC patients only in the training set of the development cohort and validation cohort using development-cohort-optimized cutoff points.

Metric (95% CI)	Development RF11	Development RF 16	Validation RF11	Validation RF16
Sensitivity	0.46 (0.18–0.73)	0.82 (0.55–1.00)	0.29 (0.13–0.50)	0.29 (0.13–0.50)
Specificity	0.66 (0.52–0.80)	0.91 (0.82–0.98)	0.86 (0.78–0.93)	0.82 (0.72–0.90)
PPV	0.25 (0.11–0.41)	0.69 (0.50–0.91)	0.41 (0.20–0.64)	0.35 (0.17–0.54)
NPV	0.83 (0.75–0.92)	0.95 (0.89–1.00)	0.78 (0.74–0.84)	0.78 (0.73–0.83)
Accuracy	0.62 (0.49–0.75)	0.89 (0.80–0.96)	0.72 (0.65–0.79)	0.69 (0.60–0.76)
F_1_ score	0.32 (0.14–0.50)	0.75 (0.56–0.91)	0.34 (0.16–0.53)	0.31 (0.15–0.48)

PPV: positive predictive value, NPV: negative predictive value, 95% confidence intervals identified with 2000 bootstrap replicates.

## Data Availability

All data required to reproduce the analyses are provided in Appendix A.

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
