# Peer review of "Validation of a Machine Learning Model to Predict Immunotherapy Response in Head and Neck Squamous Cell Carcinoma"

_cancers, 2023, doi:10.3390/cancers16010175_

Round 1
Reviewer 1 Report
Comments and Suggestions for Authors
General comments:
The article is based on a previously developed random forest machine learning model aimed to predict the efficacy of immunotherapy in general (no specific focus on a cancer type), model that is now ‘personalised’ to be applied to head and neck cancer. The paper is clearly written and generally sound.
I have a few comments that I would like the authors to consider in their revision:
1. I would expect the HPV status to be considered as a feature in a HNC cohort. Please justify the reason of not including it among your other ML parameters.
2. Add to the limitations the relatively small HNC patient cohort that was used for model training / validation.
3. While the paper focuses on the effectiveness of immunotherapy on tumour control, treatment-related side effects in a subgroup of responsive patients are known to be quite severe, with multiple organs/systems being affected, thus greatly affecting their QoL. Please comment on this aspect and on the role/ability of ML in tackling immunotherapy-related adverse effects.
Specific comments:
1. Please check the HNSCC acronym throughout the text – there are some typos that require corrections (eg. HSNCC)
Author Response
Dear reviewer,
Thank you for your thorough review of our manuscript, “Validation of a Machine-Learning Model to Predict Immunotherapy Response in Head and Neck Squamous Cell Carcinoma.” The comments you made were very constructive and have helped us to improve the manuscript.
Below, we have included a response to each comment.
Comment #1:
The article is based on a previously developed random forest machine learning model aimed to predict the efficacy of immunotherapy in general (no specific focus on a cancer type), model that is now ‘personalised’ to be applied to head and neck cancer. The paper is clearly written and generally sound.
I have a few comments that I would like the authors to consider in their revision:
- I would expect the HPV status to be considered as a feature in a HNC cohort. Please justify the reason of not including it among your other ML parameters.
Response:
We certainly agree that HPV status does convey both predictive and prognostic information in HNSCC. This study was designed to test and validate a previously described pan-cancer predictive tool, to determine if it has predictive power for patients with HNSCC. Because, in the initial development of the pan-cancer model, only 55 out of 1184 patients were HNSCC patients, HPV status was not part of the development of the initial model. As a validation study, we are limited to the features included in the initial pan-cancer model. Certainly we plan in the future work to further develop a model specific to head and neck cancer, and in such a model, we would test HPV status as a feature. This is a reasonable question and we have added an explanation of this to the Methods section (page 4, lines 150-51; page 9, lines 312-17).
Comment #2:
Add to the limitations the relatively small HNC patient cohort that was used for model training / validation.
Response:
Thank you for the suggestion. To the discussion (page 9, lines 312-315), we have added our limited sample size as a limitation of the study.
Comment #3:
While the paper focuses on the effectiveness of immunotherapy on tumour control, treatment-related side effects in a subgroup of responsive patients are known to be quite severe, with multiple organs/systems being affected, thus greatly affecting their QoL. Please comment on this aspect and on the role/ability of ML in tackling immunotherapy-related adverse effects
Response:
Thank you for the suggestion – we agree this is an important consideration. In the introduction (page 2, lines 74-77), we have added commentary on previous research which has used ML to predict immune-related adverse effects in patients being treated with immunotherapy drugs.
Comment #4:
Please check the HNSCC acronym throughout the text – there are some typos that require corrections (eg. HSNCC)
Response:
Thank you for pointing this out, and our apologies. This one typo has been corrected and we have checked the manuscript carefully.
Reviewer 2 Report
Comments and Suggestions for Authors
This study aimed to validate a random forest machine learning tool of previous study to predict the likelihood of immune checkpoint blockade (ICB) response in patients with R/M HNSCC. This study is interesting and help apply machine learning in clinical practice. This manuscript was well written, however, with several points required further clarification. This cohort also bears problems of imbalance data, therefore, the F1-score should also be considered to evaluate the performance of models. My comments to the authors as follows:
*2.1 Patient cohort: The description of this section is unclear. I read with confusion regarding what's the total number of your cohort, and how you separated your cohort into development cohort, training set, and validation cohort. Please clarify.
*2.2 Clinical Outcome and Restuls: Is it possible to add smoking status into the RF model to evaluate the prediction?
*2.5 Performance metrics and statistical analysis and Results: I noticed that there may be problems of imbalance data based on the distribution of ICB response. Could you please also use F1-score to evaluate the performance of prediction? F1-score provides more insight into the functionality of a classifier than the accuracy metric; it has been used on the assessment of models with imbalanced data in text classification. The F1-score should be added into your Result sections.
Author Response
Dear reviewer,
Thank you for your very thorough review of our manuscript, “Validation of a Machine-Learning Model to Predict Immunotherapy Response in Head and Neck Squamous Cell Carcinoma.” The comments you made were very constructive and were helpful in improving the manuscript.
Below, we have included a response to all of your suggestions.
Comment #1
This study aimed to validate a random forest machine learning tool of previous study to predict the likelihood of immune checkpoint blockade (ICB) response in patients with R/M HNSCC. This study is interesting and help apply machine learning in clinical practice. This manuscript was well written, however, with several points required further clarification. This cohort also bears problems of imbalance data, therefore, the F1-score should also be considered to evaluate the performance of models. My comments to the authors as follows:
Patient cohort: The description of this section is unclear. I read with confusion regarding what's the total number of your cohort, and how you separated your cohort into development cohort, training set, and validation cohort. Please clarify.
Response:
Thank you for the comment – we agree this was not well described and is important for readers to be able to follow. This section has now been reworded to improve clarity. The development cohort (1479 patients) was compiled during the development of the model and split into the training set and test set in an 80:20 ratio as described by Chowell et al. The validation cohort was compiled specifically for this study, and includes 96 HNSCC patients which were not present in the development cohort. This has now been clarified in several locations in the Methods (page 3, lines 100-113).
Comment #2
*2.2 Clinical Outcome and Results: Is it possible to add smoking status into the RF model to evaluate the prediction?
Response:
We certainly agree with the reviewer that smoking status is an important prognostic factor, and may perhaps also be an important predictive factor for immunotherapy response, in patients with HNSCC. For this particular study, which was designed as a validation study of a previously described pan-cancer predictive tool, we were limited to validating the existing tool. Therefore, we did not re-train a new head and neck cancer-specific model with additional features. We do plan to accomplish this in future work and certainly would test smoking history as a potentially very informative feature. We have added explanation of this to the manuscript in a few locations (page 4, lines 150-51; page 9, lines 312-17).
Comment #2
Performance metrics and statistical analysis and Results: I noticed that there may be problems of imbalance data based on the distribution of ICB response. Could you please also use F1-score to evaluate the performance of prediction? F1-score provides more insight into the functionality of a classifier than the accuracy metric; it has been used on the assessment of models with imbalanced data in text classification. The F1-score should be added into your Result sections.
Response:
We thank you for this very useful suggestion and of course, agree. The F1 score has been added to the results (page 7, lines 239-240, Table 2).
Round 2
Reviewer 2 Report
Comments and Suggestions for Authors
The authors had addressed my concerns.